# Measurement of Fitness and Predatory Ability of Four Predatory Mite Species in Tibetan Plateau under Laboratory Conditions

**DOI:** 10.3390/insects15020119

**Published:** 2024-02-06

**Authors:** Dong Xiang, Zhen Wang, Long Xu, Yunchao Wang, Huanhuan Zhang, Kun Yang

**Affiliations:** 1Institute of Vegetable, Tibet Academy of Agricultural and Animal Husbandry Sciences, Lhasa 850032, China; xiangd666@126.com (D.X.); 15895980277@163.com (Z.W.); 2Qingdao Agricultural Administrative Law Enforcement Detachment, Qingdao 266000, China; 190361657@163.com; 3College of Biology and Agriculture, Zunyi Normal University, Zunyi 563006, China; wyc801suo@163.com; 4Shandong Engineering Research Center for Environment-Friendly Agricultural Pest Management, College of Plant Health and Medicine, Qingdao Agricultural University, Qingdao 266109, China

**Keywords:** fitness, predatory mites, biological control, spider mite, Tibetan Plateau

## Abstract

**Simple Summary:**

Two-spotted spider mites (*Tetranychus urticae* Koch, TSSMs) greatly harm vegetables and other crops in Tibet, but predatory mites are potential effective biological agents against them. Herein, we measure the fitness and predatory abilities of four predatory mite species in the Tibetan Plateau and confirm that the predatory mite *Amblyseius swirskii* (Athias-Henriot) (Acari: Phytoseiidae) had the highest fecundity, the highest pre-adult survival rate, and the highest predation capacity toward adult TSSMs at 15 d post-release. The results imply that *A. swirskii* is more effective for the control of TSSMs in laboratory conditions and can be viewed as an effective biological control agent candidate against TSSMs in Tibet.

**Abstract:**

Predatory mites are biological control agents used in many countries against various vegetable pests, particularly spider mites. Despite the significant presence of predatory mites in the Tibetan plateau, there is limited research on their potential against spider mites in the area. This study investigated the fitness parameters and performance against TSSM of four predatory, including *Amblyseius swirskii* (Athias-Henriot) and three species from the genus *Neoseiulus* (*Neoseiulus californicus* (McGregor), *Neoseiulus barkeri* (Hughes), and *Neoseiulus cucumeris* (Oudemans)), originally collected from fields in the Tibetan Plateau. Compared to the other three predatory species, *A. swirskii* exhibited the highest fecundity (11.60 ± 0.34) and the highest pre-adult survival rate (83.33 ± 3.33%). Since their juvenile survival rate (SR) was extremely low (13.33% ± 5.77%), most *N. barkeri* nymphs died before emergence. Compared to the other three predatory mites, *A. swirskii* showed the highest predation capacity against adult TSSMs at 15 d post-release (14.28 ± 2.24). Based on the results, *A. swirskii* was the most effective, and *N. barkeri* was the least effective in controlling two-spotted mites in the Tibetan Plateau among the four species tested in this study. Collectively, these findings imply notable advantages in employing *A. swirskii* for controlling two-spotted mites in the Tibetan Plateau. This study informs the development of a feasible biological control method based on suitable predatory mite species to manage TSSMs in the Tibetan Plateau.

## 1. Introduction

In all mite families, the Tetranychidae family, which encompasses > 1300 species of spider mites, holds a significant position regarding its impact on agricultural production [1,2]. The two-spotted spider mite (TSSM) is the most widely distributed and destructive spider mite species worldwide, feeding on more than 1150 host plants, including many important crops, such as corn, soybean, tomato, and cucumber, among others [1,3]. Furthermore, multiple TSSM populations have strongly resisted major acaricides and insecticides in various regions [4]. Since 1983, TSSMs have been widespread in most regions in China, and they are currently the dominant spider mite species [5,6]. Specifically, in the Tibetan Autonomous Region of China, TSSMs have been reported to harm numerous vegetable crops, including soybean, tomato, and pepper. TSSM-related damage is mostly observed in greenhouses as opposed to fields in Tibet and is also difficult to control [7]. In our previous study, the biocontrol effectiveness of four predatory species (*Amblyseius swirskii*, *Neoseiulus californicus*, *Neoseiulus cucumeris*, and *Neoseiulus barkeri*) on TSSMs was measured [8]. However, more studies should be carried out on the biological management of spider mites in Tibet, the world’s highest plateau.

As a polyphagous predator, *N. californicus* mainly feeds on Tetranychid spider mites (especially TSSMs and other spider mites such as *T. evansi*) [9,10,11]. It can also consume various small insects, including other mite species or thrips; hence, McMurtry and Croft classified it as a type II specialist predatory mite (selective predators of Tetranychid mites) [12].

On the other hand, predatory mites of the genus *Amblyseius*, which are popular for their tendency to feed on various prey, including a multitude of mites from many families (such as Tetranychidae, Eriophyidae, Pyroglyphidae, and so on) and many other arthropods, including enormous pest insects (such as whiteflies, thrips, nematodes, and so on) [12,13], have always been considered generalist predators, and hence, were listed as subtype III-b of type III lifestyle predators by McMurtry et al. [13]. The species *N. californicus* and other predators also belong to the genus *Amblyseius*; the impact of highland conditions on them has received little research attention, and their predatory capacity remains largely unknown.

Chemical acaricides and insecticides are the primary methods for controlling TSSMs. Over the past few decades, TSSMs have rapidly developed a global resistance to these chemical agents, especially in China [14,15,16,17]. In Tibet, the world’s largest highland, chemical pesticides are widely applied to combat pests, and the resulting significant residues became an issue of concern [15]. Interestingly, no treatments other than chemical control are used in Tibet to effectively prevent the harm caused by two-spotted mites. Predatory mites are rarely used in Tibet, and pertinent research has also been extremely limited, even in our studies, although they are important biological control agents against many pest species, including spider mites; hence, the fitness and predatory capacity of many predatory mites on TSSMs in this region remain unclear [8,18,19].

With an average elevation of >4000 m above sea level, the Tibetan Plateau is the highest of China’s three major landform ladders. It is characterized by low temperatures and hypoxia, the most substantial threats to the lives of arthropods and other animals in the area [20,21,22,23,24]. Invertebrates often develop various useful strategies to combat adverse highland conditions. For example, to combat hypoxia conditions (2% O_2_), the bean weevil increases its synthesis of metabolites, including carbohydrates, amino acids, and organic acids [25]. Furthermore, since the cytochrome oxidase activity is higher in Tibetan locusts, they show a higher hypoxia tolerance than lowland locusts [26], while the high-altitude condition has significant effects on the fitness and transcriptome of the diamondback moth (*Plutella xylostella* (Linnaeus) (Lepidoptera: Plutellidae)) [27]. Additionally, as a means of surviving the high-latitude environment, the pollinator *Bombus pyrosoma* (Morawitz) (Hymenoptera: Apidae) upregulates energy metabolism genes [28]. Another study showed that with the changing climate, range shifts are already occurring in animals, especially for ectotherms and high-latitude species [29]. Despite these useful insights, there have been no recent explorations into the effects of highland conditions on the fitness and prey capacity of predatory mites.

In our previous study, we measured the biological control effectiveness of four predatory mite species, including the species *Amblyseius swirskii* and three species from the genus *Neoseiulus* (*N. californicus*, *N. barkeri*, and *N. cucumeris*), against TSSMs and *T. cinnabarinus* in greenhouse conditions [8]. Although *Phytoseiulus persimilis* was considered the most effective predatory mite for controlling spider mites [10], the bio-invasion problem should be considered as there are no reports of *P. persimilis* in Tibet; hence, in this experiment, the species *P. persimilis* was excluded. Since the eco-environment is fragile and biodiversity is low in Tibet, and chemical pesticides would cause worse damage in Tibet than in low-altitude areas, biocontrol in Tibet is critical [7,8,19].

## 2. Materials and Methods

### 2.1. Establishment of TSSMs and Predatory Mite Populations

The TSSM population was collected from soybean leaves in the solar greenhouse of the Lhasa Agricultural Science and Technology Park (roughly E91°07′, N29°38′). Spider mite species identification was confirmed by COI sequencing [30]. After introducing the spider mites into the laboratory, the isofemale line of TSSMs was established and grown on the leaves of kidney beans (*Phaseolus vulgaris* L.) in the laboratory under controlled conditions (25 ± 1 °C, 60% humidity, and under 16 h light/8 h dark conditions), which are similar rearing conditions to those of greenhouses in Tibet. The TSSM population was used as the target or objective pest mite to determine the prey capacity of the predatory mites.

In this study, four predatory mite species, including *A. swirskii*, *Neoseiulus barkeri* (Hughes), *Neoseiulus cucumeris* (Oudemans), and *N. californicus* were examined. All predatory mite species were purchased from Fujian Yanxuan Biological Control Technology Company (Fuzhou, China) and categorized based on three characteristics: their spermatheca, abdominal shield, and dorsal shield [31].

### 2.2. Fitness Experiment for the Four Predatory Mite Species

The fitness of all four predatory species was assessed after rearing in highland conditions in Lhasa, Tibet Autonomous Region, China (roughly E91°07′, N29°38′) for more than ten generations (approximately six months). The rearing conditions were as follows: (1) temperatures of 25.5 ± 1 °C; (2) relative humidity (RH) of 60 ± 5%; (3) a photoperiod of 16:8 h (L/D); and (4) a mean altitude of 3642 m above sea level. First, thirty new 1-day-old emerged female adults and thirty new 1-day-old emerged male adults were selected for each mite species to oviposit for 4 h in one dish (9 cm). Following that, 30 eggs of each predatory species were randomly selected to conduct the subsequent test, and all eggs of one predatory species was placed in a separate dish with a kidney bean leaf (about 7 cm) as a replicate, with each mite species having 30 replicates. In each dish, five TSSM eggs and five TSSM nymphs (mixed age stages) were added every five days as a diet. To calculate the juvenile survival and developmental rates of the different predatory mite species, the stages of all predatory mites were monitored and recorded every 12 h. Ten predatory mites were regarded as a replicate for the juvenile survival calculation. Each mite species experiment was performed in triplicate.

Once a female predatory mite emerged, one newly emerged male adult was introduced to mate with one female adult. Ten female adults of each mite species were randomly selected each day to record fecundity. Given that the pre-adult survival rate (SR) of *N. barkeri* was extremely low, there were not enough adults to record the fecundity, so ten 1-day-old female *N. barkeri* adults were selected from other leaf dishes to record the fecundity. The fecundity and hatchability data were first tested for normality (Kolmogorov–Smirnov test) and then the homogeneity of group variances (Levene’s test). Subsequently, one-way ANOVA analysis (SPSS 21.0, IBM SPSS statistics, Chicago, IL, USA) was employed to determine the variance of fecundity and the developmental rates among all four predatory mites. Given that the SRs did not follow a normal distribution, they were analyzed by a Kruskal–Wallis test, and multiple comparisons were performed using Dunn’s test with Bonferroni correction (SPSS 21.0, IBM SPSS statistics, Chicago, IL, USA). The Cox proportional hazard model was used to compare the survival curves of each predatory mite’s female adults (SPSS 21.0, IBM SPSS statistics, Chicago, IL, USA).

### 2.3. Functional Response

The functional response experiments on the ability of the four predatory mite species to prey on TSSMs (different stages) were conducted under 25 °C conditions. First, the equivalent numbers of TSSMs at three variant stages (egg, nymph, and adult) were selected at six density gradients (1, 3, 5, 7, 9, and 11), and the same numbers of TSSM eggs, nymphs, and adults were all placed in a dish, while one 1-day-old female adult predatory mite (newly emerged, which was nongravid) was placed in the same dish after 24 h of starvation. After 24 h, the predation activities of the predatory mites at different stages of the TSSM were recorded. The preying of predatory mites on different stages of the TSSM was measured using a predator–prey model, specifically the Holling type II functional response [32]:$$N_{a}=\frac{ɑ*T*N}{1+T_{h}*ɑ*N}$$
where *N_a_* = the number of TSSM mites preyed on, ɑ = the attack rate (proportion of prey captured by each predator per unit of searching time), *T_h_* = the handling time (the time it takes for predatory mites to identify, kill, and consume the TSSM), *N* = the prey density, and *T* = the time it takes for predators to find the prey (in this test, *T* is 1 d).

### 2.4. Predator Interactions at Different Predatory Mite Densities

The predatory abilities of each biological control agent at various densities were measured. In this study, six predatory mite density gradients were used (1, 3, 5, 7, 9, and 11). Each mite species was prepared for each density, and these combinations were replicated three times. For every replicate, forty TSSM female adults were first put in a leaf dish. One selected density of 1-day-old female adults from a single predatory species was then placed in the same dish, and the preyed number of different stages of the TSSMs was recorded after 24 h. Predator interference analysis was performed using Watt’s model of the effect of the densities of attacked and attacking species on the number of the attacked [33]:$$A={ɑX}^{b}$$
where *A* = the number of TSSM female adults preyed on; the predatory ability of mites was calculated with *A* to divide the predatory days (1 d or 15 d). ɑ = the attack rate without competition, *X* = the density of predators per leaf dish, and *b* = the intraspecific competition parameter. *a* and *b* are all coefficients in this equation, and they are calculated by: ln (*A*) = ln(*a*) + (−*b*) ln(*X*).

The data on the predatory capacity, handling time, attack coefficient, and max predatory amount were tested for normality (Kolmogorov–Smirnov test) and the homogeneity of group variances (Levene’s test) if they followed a normal distribution. Then, one-way ANOVA analysis (SPSS 21.0, IBM SPSS statistics, Chicago, IL, USA) was performed to determine the variance among all four predatory mites. On the other hand, the data were subjected to a Kruskal–Wallis test, and multiple comparisons were performed using Dunn’s test with Bonferroni correction if they followed a non-normal distribution (SPSS 21.0, IBM SPSS statistics, Chicago, IL, USA).

## 3. Results

### 3.1. Fitness of Four Predatory Mites at the Tibetan Plateau

One-way ANOVA was used as all the mites’ developmental time data followed a normal distribution. When the development times of the four predatory mites from eggs to emergence were compared, *N. barkeri* (14.50 ± 0.61 d) and *N. cucumeris* (13.54 ± 0.15 d) had a significantly longer juvenile period in the Tibet area. On the other hand, *N. californicus* had the fastest developmental rate (9.50 ± 0.16 d) (*F*_3,70_ = 141.38, *p* < 0.001), implying that it had an applicability advantage over the remaining three predatory mite species grown in Tibet (Figure 1).

In terms of the specific developmental stages, *A. swirskii* and *N. californicus* had similar developmental rates for the egg period and larva stage, showing that they could spend non-significantly different periods to complete both growth stages. On the other hand, *N. barkeri* and *N. cucumeris* took significantly longer periods to complete the two growth stages (*p* < 0.05) (Appendix A). Additionally, compared to the other three mite species, *N. californicus* had the fastest developmental rate for the protonymph stage. On the other hand, although the growth time of *N. californicus* during the deutonymph stage was significantly shorter than that of *N. barkeri* and *N. cucumeris*, it was comparable to that of *A. swirskii* (Appendix A).

### 3.2. Predatory Mite Fecundity

The egg counts of all predatory mites were determined per female. Consistent with the developmental time data, all fecundity data followed a normal distribution. According to the results, *A. swirskii* (11.60 ± 0.65) had a significantly higher fecundity compared to the other four predatory species, while *A. barkeri* had the lowest fecundity (9.60 ± 0.34) (*F*_3,36_ = 2.57, *p* < 0.05). Compared to those of *A. swirskii* or *N. barberi,* the fecundities of *N. cucumeris* (10.30 ± 0.63) and *N. californicus* (10.10 ± 0.43) were non-significantly different (Figure 2).

### 3.3. Survival Rates at the Tibetan Plateau

During rearing in the Tibet plateau, *A. swirskii* had the highest pre-adult SR (83.33% ± 5.77%), while *N. barberi* had the lowest SR (13.33% ± 5.77%), significantly lower than those of the other three predatory species (*F*_3,12_ = 7.37, *p* < 0.05). At the same time, the pre-adult SRs of *N. cucumeris* and *N. californicus* were not significantly different compared to that of *A. swirskii* (Figure 3).

### 3.4. Longevity

Each species’ longevity from mite emergence to death was measured to further clarify the influence of highland conditions on the life spans of the four predatory species. According to the results, there were no significant differences among all four predatory species (*p* = 0.44, Cox proportional hazard model), but *N. barkeri* had the greatest longevity (median survival = 38 d) (Figure 4).

### 3.5. Evaluation of the Predatory Capacity of Predatory Mites

The prey capacities of the four predatory mite species with different predator densities were measured at 1 d and 15 d post-release. At 1 d post-release, most species exhibited the highest predatory capacity on TSSMs at density 5. Specifically, *A. swirskii* had the highest predatory capacity at density 5 (11.20 ± 1.30), and two other species, *N. barkeri* (10.20 ± 1.30) and *N. californicus* (8.20 ± 0.84,) also had a high predatory capacity at the same density. On the other hand, *N. cucumeris* had the highest predatory capacity on spider mites at density 1 (9.40 ± 2.07) (Figure 5).

At 15 d post-release, the highest predatory capacity was observed at density 9 for all mites except *A. swirskii*, which had the highest capacity at density 7 (23.00 ± 1.87). Compared to *N. cucumeris* and *N. barkeri, N. californicus* had a higher predatory capacity at density 9 (22.60 ± 1.82) (Figure 6). At the same time, the data in Figure 5 and Figure 6 successfully fit into the Holling II disc equation. The attack coefficient (α), handling time (Th (d)), and maximum predation capacity (1/Th) results are all summarized in Appendix A (1 d post-release) and Appendix A (15 d post-release). Notably, the predatory mites’ prey capacity (preyed number of TSSMs per day) was always significantly higher at 15 d post-release than at 1 d post-release (Figure 7).

### 3.6. Functional Responses at Different Developmental Stages

The functional response experimental results were subjected to the Holling type II functional response analysis to obtain the functional response models of each predatory mite species’ prey capacities at different stages of the TSSM. At 1 d post-release, the data on the predatory capacity of all four predatory mite species did not follow a normal distribution. Consequently, nonparametric analysis was employed, showing that the predatory capacities of all four predatory mite species were significantly different (*F*_11,36_ = 23.01, *p* < 0.05). However, multiple comparisons through Dunn’s tests with Bonferroni correction showed no significant difference between any two groups, with *N. cucumeris* (10.14 ± 3.41) having the highest predatory capacity on adult TSSMs, and *A. swirskii* (7.58 ± 3.25) also having a high predatory capacity on adult TSSMs (Figure 8A). Additionally, the four predatory mite species’ data on handling time and attack coefficient did not follow a normal distribution. Although no statistically significant difference was detected among the four species in handling time (*F*_11,36_ = 11.00, *p* = 0.29) (Figure 8B), there was a statistically significant difference between them in the attack coefficient data (*F*_11,36_ = 27.84, *p* < 0.01) (Figure 8C), and Dunn’s tests with Bonferroni correction for multiple comparisons revealed no statistically significant difference in the attack coefficient data (Figure 8C). The max predatory amount data of all four predatory mite species followed a normal distribution, but with no significant difference [*F*_11,24_ = 1.272, *p* = 0.298] (Figure 8D).

At 15 d post-release, the predatory mites’ data on predatory capacity, handling time, and max predatory amount all followed a normal distribution. A statistically significant difference was found in all mites’ predatory capacity [*F*_11,24_ = 7.47, *p* < 0.001]. Specifically, *A. swirskii* (14.28 ± 2.24) had the highest predatory capacity on adult TSSMs, *with N. cucumeris* (11.62 ± 1.48) also having a relatively high predatory capacity on adult TSSMs. On the other hand, *N. barkeri* had the lowest predatory capacity on various stages of TSSMs (Figure 9A). No statistically significant difference was found in the handling time [*F*_11,24_ = 1.245, *p* = 0.312] (Figure 9B). The attack coefficient data did not follow a normal distribution, and significant differences were found in the attack coefficients among the four predatory mite species (*F*_11,36_ = 20.18, *p* < 0.05). On the other hand, Dunn’s tests with Bonferroni correction for multiple comparisons found no statistically significant difference in the attack coefficient data (Figure 9C). There was no significant difference detected in the mite species’ max predatory amount data (Figure 9D).

Multiple linear regression analysis was used to analyze the correlation between the attack rate (ɑ) and the handling time (*T_h_*) with predatory mite species, TSSM growth stages, and the predatory period, and the results show that both the predatory mite species (*p* < 0.001) and the TSSM growth stages (*p* < 0.001) significantly influenced the attack rate (ɑ) (Appendix A), and the predatory mite species (*p* < 0.01), TSSM growth stages (*p* < 0.01), and predatory period (*p* < 0.05) significantly affected the handling time (*T_h_*) of predators (Appendix A).

## 4. Discussion

In this study, the fitness and prey capacity of four predatory mite species (*A. swirskii*, *N. barkeri*, *N. cucumeris,* and *N. californicus*) were assessed. According to the results, *A. swirskii* had the highest fecundity and pre-adult SR, *N. californicus* took the least time to transition from the egg stage to emergence, and *N. barkeri* could not survive in highland conditions, as most of its nymphs died before emergence. Furthermore, *A. swirskii* had the highest prey capacity on TSSMs compared to the other three predatory mite species based on their functional response. Our findings collectively imply that among the four tested groups, *A. swirskii* showed potential to be used as a biological control agent against TSSMs in the Tibetan plateau.

Multiple studies reported that high-altitude conditions could significantly influence arthropods’ development, morphological characteristics, and body size, among many other aspects [34,35,36,37,38]. For example, a significant link was found between insects’ body size and altitude in a study on wing-reduced stonefly in New Zealand [39]. Furthermore, insects’ fitness was found to decrease with increasing elevation, as demonstrated by the reduced fecundity of the willow leaf beetle (*Chrysomela aeneicollis* Schaeffer (Coleoptera: Chrysomelidae)) under high-altitude conditions [35]. Herein, we confirmed that a high altitude significantly adversely affected *N. barkeri*, resulting in an extremely low SR before emergence, whereas the other three predator species had a better SR as they all had a relatively better fitness, especially the *A. swirskii* female adult mites, which had the highest fecundity and pre-adult SR.

*A. swirskii* is one of the most successful biocontrol agents globally, as it can prey on many severe pests and is easy to rear [40]. Furthermore, another study in China indicated that *A. swirskii* had an advantage in the predation of whiteflies over other two predatory mite species: *A. orientalis* and *N. californicus* [41]. Our previous study about the predatory capacity of four predatory mite species on peach trees in a greenhouse in Tibet revealed the advantage of *A. swirskii* on biocontrol [8]. These results are consistent with our findings, as *A. swirskii* showed a significantly higher predatory capacity on TSSMs than *N. californicus* at 15 d post-release (Figure 9A), and its fitness in Tibet was comparable to that of similar strains reared under normal conditions [42]. As for *N. californicus*, our results indicate that although most fitness parameters, including developmental time, pre-adult SR, and longevity in Tibet were comparable to those of mites reared in low-altitude conditions [43,44,45], its fecundity in Tibet was significantly lower compared to that under relatively lower elevation laboratory conditions, where the rearing conditions of *N. californicus* at low altitude was similar to the present study (25 ± 1 °C, 60 ± 5% RH and a photoperiod of 16:8 h) [37]. This finding is unsurprising as, just like other arthropods, mites’ fitness could be influenced by many abiotic and biotic factors [43,46,47].

Phytoseiid mites, including *N. californicus*, are widely distributed biological agents for controlling spider mites. They have been used for decades and have attracted substantial attention given their satisfactory capacity to control the harm caused by TSSMs [37,39,48,49]. Notably, the prey capacity of Phytoseiid mites on spider mites varies under various conditions, including different temperature ranges [48,49], biotic factors, such as host plants, and prey quality and prey densities [20]. Here, we elucidated the influence of high altitudes on the prey capacities of four predatory mite species and deduced that compared to the other three predators, *A. swirskii* had the highest predation capacity on adult TSSMs at 15 d post-release (14.28 ± 2.24). The predatory capacities of all mite species at 15 d were significantly higher than at 1 d, which shows a increasing capacity with time, indicating an applicability advantage over the other three species in controlling two-spotted mites in the Tibetan Plateau.

Overall, we found that *A. swirskii* had a relatively higher fitness and better predatory capacity than the other predatory mites for controlling TSSMs in a highland area. Furthermore, the density of nine predatory mites per plant was the most effective in controlling spider mites, and *N. barkeri* was an unsuitable biological control agent under highland conditions. Our findings could inform the development of feasible biological control agents for managing TSSMs in the Tibetan Plateau using suitable predatory mite species. They could also be useful in delaying the emergence or increase in resistance to chemical pesticides in two-spotted mites in the Tibet plateau.

## Figures and Tables

**Figure 1 insects-15-00119-f001:**
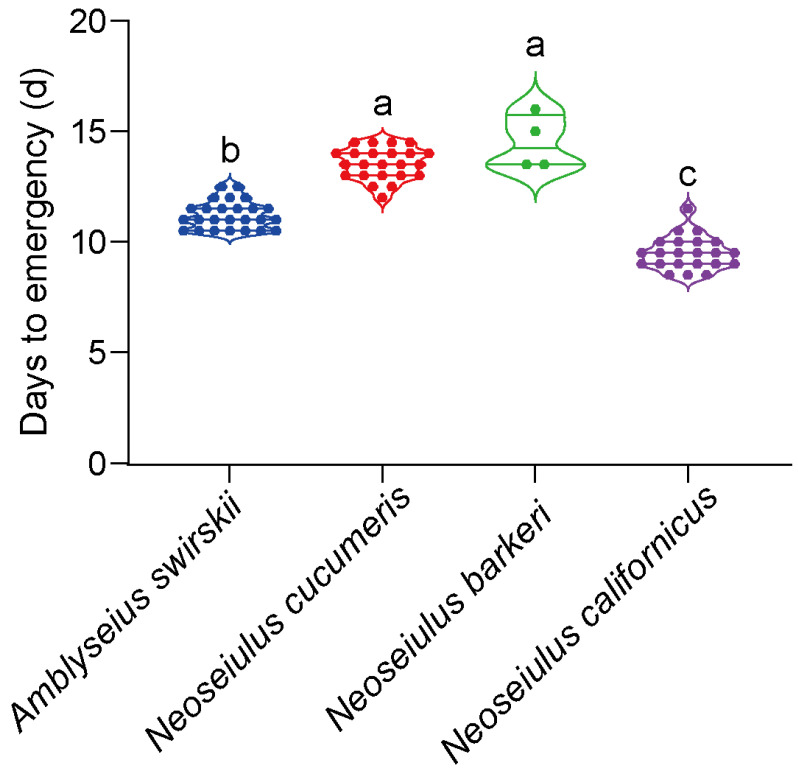
Developmental times of four predatory mite species in Tibetan Plateau. The same letters above the columns represent homogeneous groups in post-hoc tests (*p* > 0.05) following an ANOVA, different letters above the columns represent significant differences between variables in post-hoc tests (*p* < 0.05) following ANOVA. Different colors mean different predatory species.

**Figure 2 insects-15-00119-f002:**
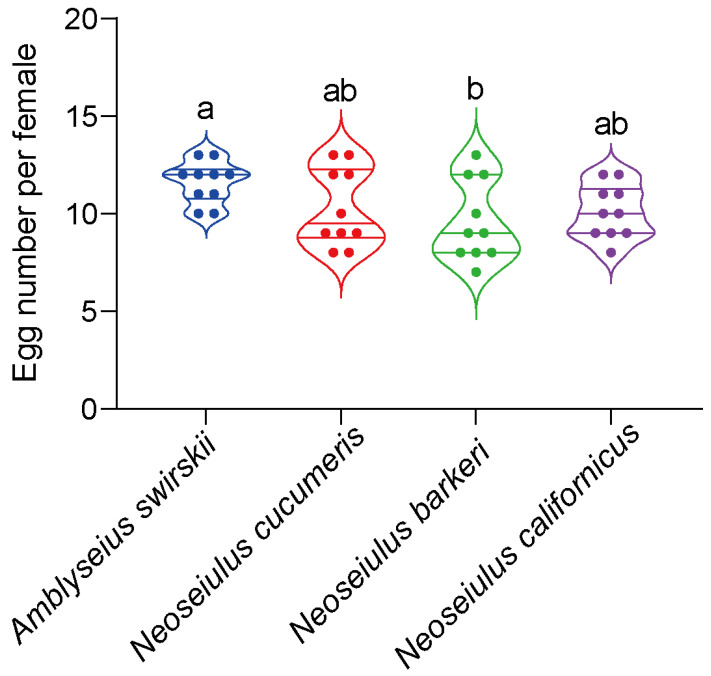
Fecundity of four predatory mite species in the Tibetan Plateau. The same letters above the columns represent homogeneous groups in post-hoc tests (*p* > 0.05) following ANOVA, different letters above the columns represent significant differences between variables in post-hoc tests (*p* < 0.05) following ANOVA. Different colors mean different predatory species.

**Figure 3 insects-15-00119-f003:**
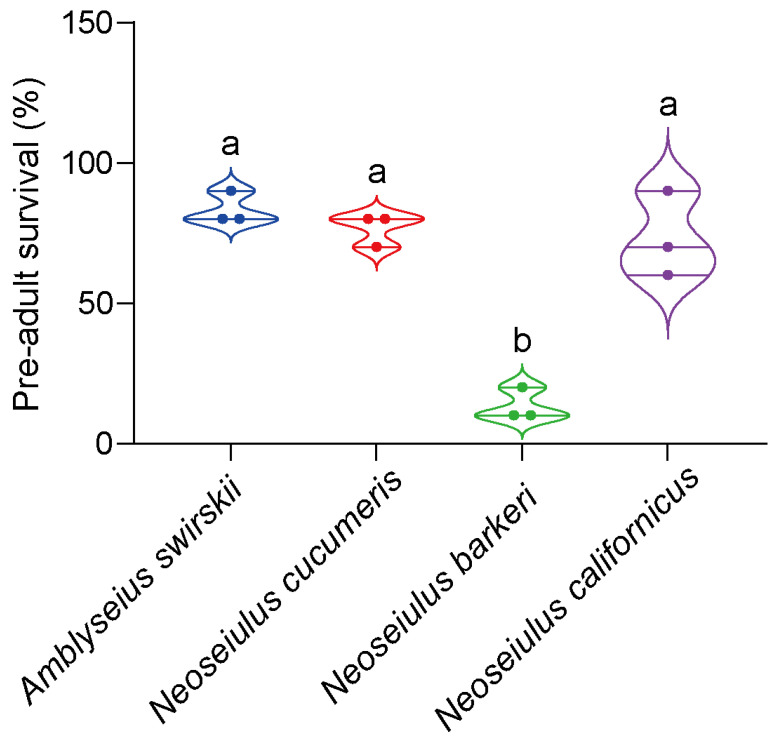
Pre-adult survival rates of four predatory mite species in Tibetan Plateau. The same letters above the columns represent homogeneous groups in post-hoc tests (*p* > 0.05), different letters above the columns represent significant differences between variables in post-hoc tests (*p* < 0.05) followed by Kruskal–Wallis tests and Dunn’s tests with Bonferroni correction for multiple comparisons. Different colors mean different predatory species.

**Figure 4 insects-15-00119-f004:**
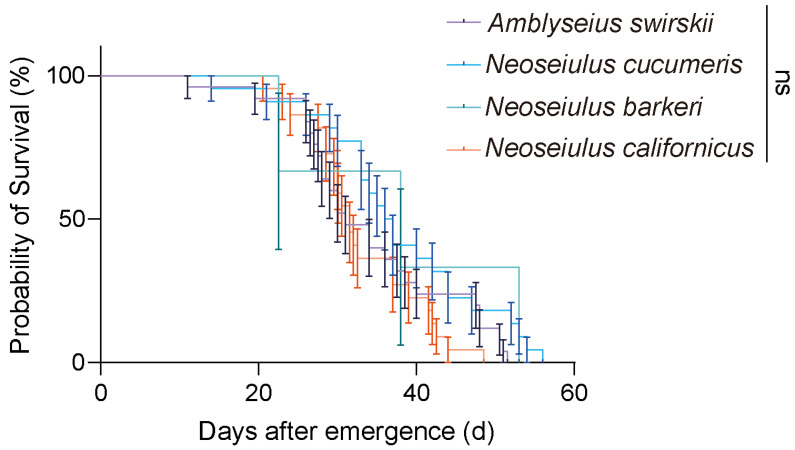
Longevity of four predatory mite species in the Tibetan Plateau. Survival curves for individual mites were compared using the Cox proportional hazard model. Error bar stands for the standard deviation. Different colors mean different predatory species. ns, not significant.

**Figure 5 insects-15-00119-f005:**
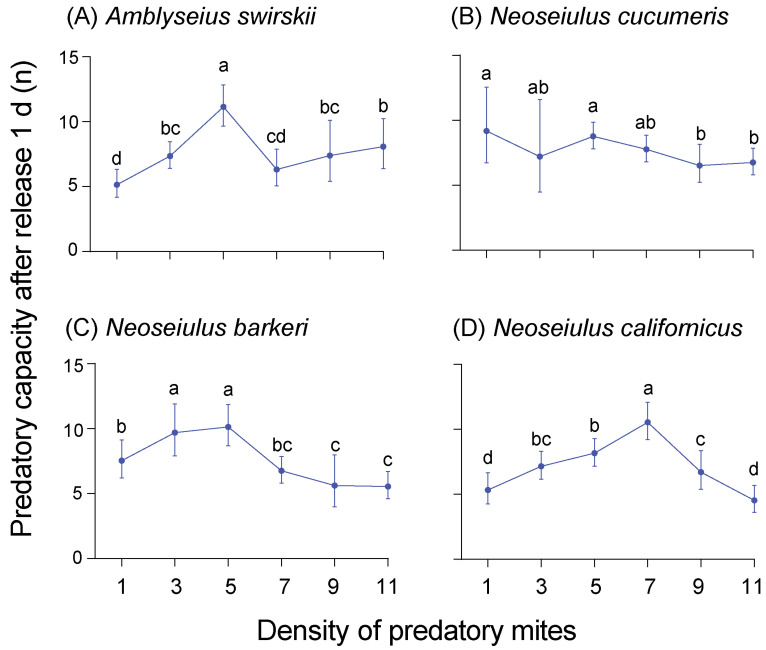
Predatory capacity (preyed spider mites per predator) after release of 1 d of four predatory mite species in Tibetan Plateau, including *Amblyseius swirskii* (**A**), *Neoseiulus cucumeris* (**B**), *Neoseiulus barkeri* (**C**) and *Neoseiulus californicus* (**D**). The same letters above the columns represent homogeneous groups in post-hoc tests (*p* > 0.05) following ANOVA, different letters above the columns represent significant differences between variables in post-hoc tests (*p* < 0.05) following ANOVA.

**Figure 6 insects-15-00119-f006:**
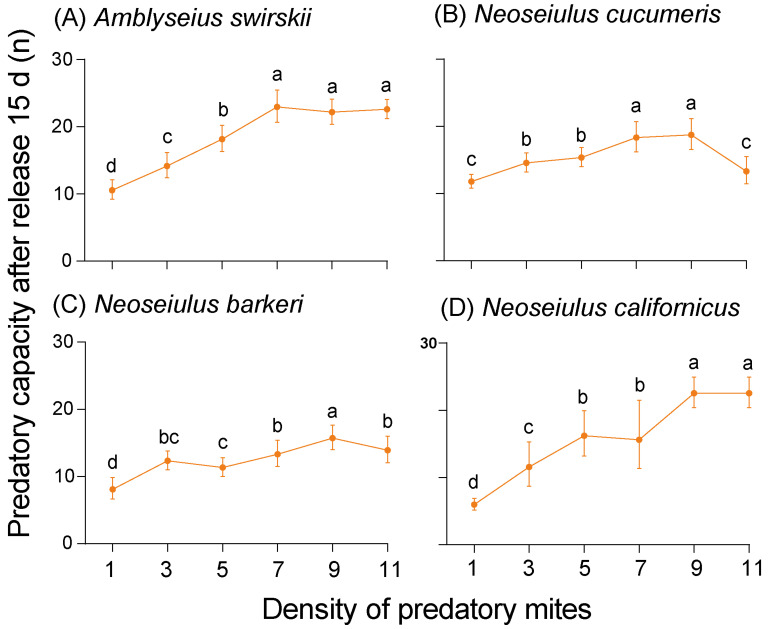
Predatory capacity (preyed spider mites per predator) after release of 15 d of four predatory mite species in the Tibetan Plateau, including *Amblyseius swirskii* (**A**), *Neoseiulus cucumeris* (**B**), *Neoseiulus barkeri* (**C**) and *Neoseiulus californicus* (**D**). The same letters above the columns represent homogeneous groups in post-hoc tests (*p* > 0.05) following ANOVA, different letters above the columns represent significant differences between variables in post-hoc tests (*p* < 0.05) following ANOVA.

**Figure 7 insects-15-00119-f007:**
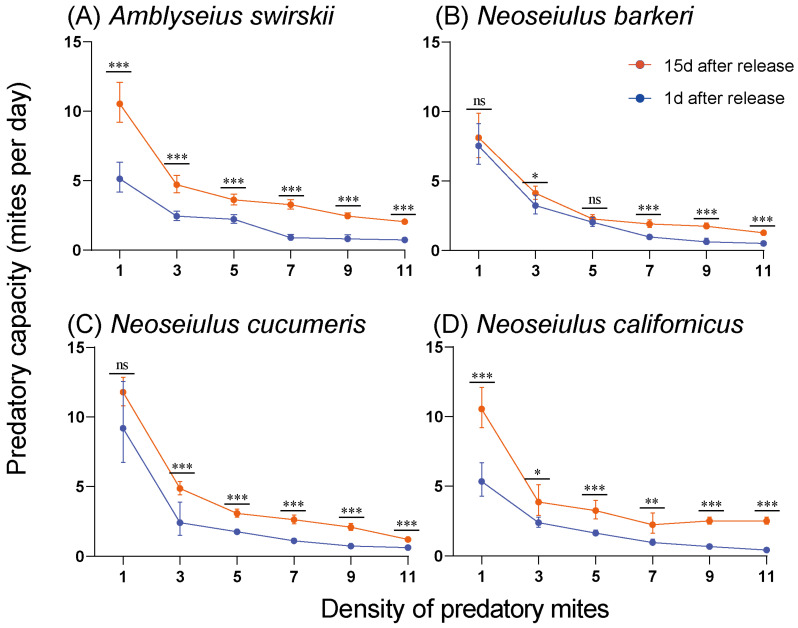
Predatory capacity (preyed spider mites per predator) between release of 1 d and 15 d of four predatory mite species in the Tibetan Plateau, including *Amblyseius swirskii* (**A**), *Neoseiulus barkeri* (**B**), *Neoseiulus cucumeris* (**C**) and *Neoseiulus californicus* (**D**). Asterisks above points indicate statistically significant differences (Mann–Whitney U-test, * *p* < 0.05, ** *p* < 0.01, *** *p* < 0.001, ns, not significant).

**Figure 8 insects-15-00119-f008:**
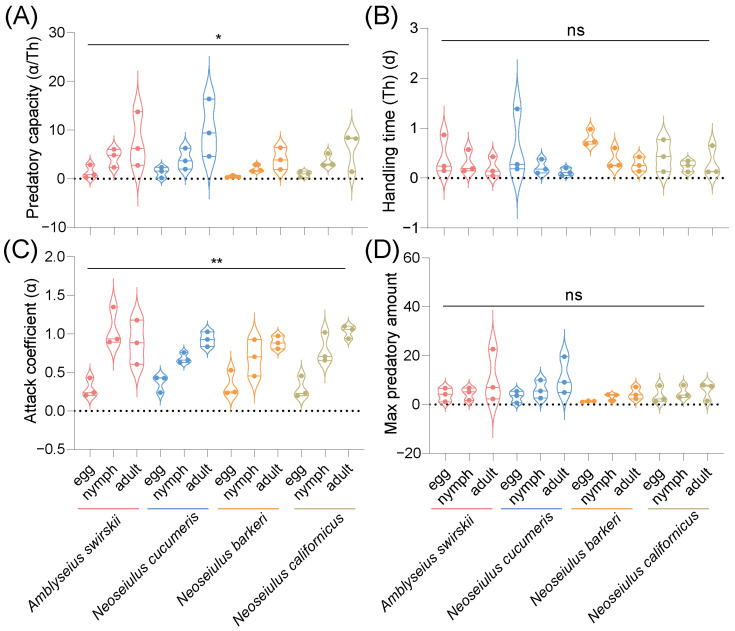
Predatory capacity (**A**), handling time (**B**), attack coefficient (**C**), and max predatory amount (**D**) after release of 1 d of four predatory mite species in the Tibetan Plateau. The max predatory amount data were analyzed by one-way ANOVA, SPSS 21.0, while the data of the predatory capacity, handling time, and attack coefficient were analyzed by Kruskal–Wallis tests, SPSS 21.0 (* *p* < 0.05, ** *p* < 0.01, ns, not significant). α = attack rate (proportion of prey captured by each predator per unit of searching time), *T_h_* = handling time (the time it takes for predatory mites to identify, kill, and consume TSSMs).

**Figure 9 insects-15-00119-f009:**
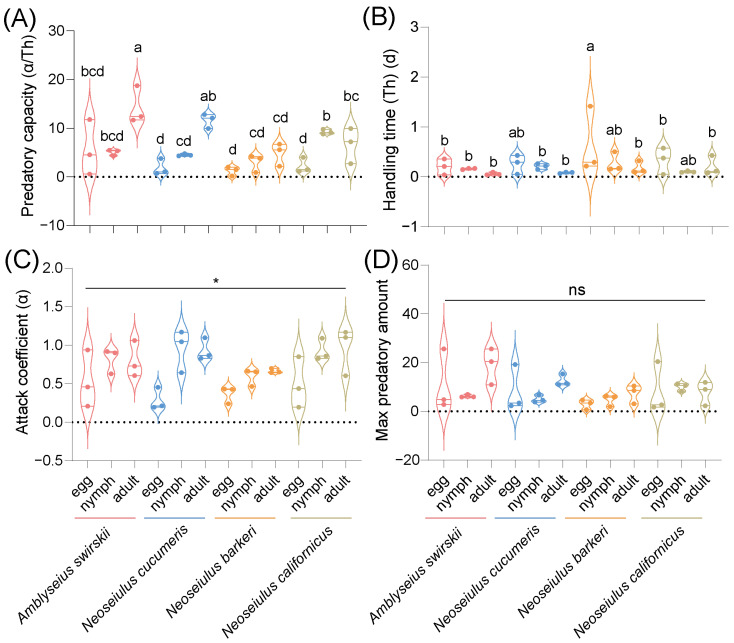
Predatory capacity (**A**), handling time (**B**), attack coefficient (**C**), and max predatory amount (**D**) after release of 15 d of four predatory mite species in the Tibetan Plateau. Results are shown as box and whiskers; the same letters above the columns represent homogeneous groups in post-hoc tests (*p* > 0.05) following ANOVA, different letters above the columns represent significant differences between variables in post-hoc tests (*p* < 0.05) following ANOVA except for the attack coefficient, which was analyzed by Kruskal–Wallis tests, SPSS 21.0 (* *p* < 0.05). ɑ = attack rate (proportion of prey captured by each predator per unit of searching time), *T_h_* = handling time (the time it takes for predatory mites to identify, kill, and consume TSSMs). Different colors mean different predatory species.

## Data Availability

The data presented in this study are available within the article and the Appendix A.

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
