# Peer review of "Measurement of Fitness and Predatory Ability of Four Predatory Mite Species in Tibetan Plateau under Laboratory Conditions"

_insects, 2024, doi:10.3390/insects15020119_

Round 1
Reviewer 1 Report
Comments and Suggestions for Authors
My comments for the authors are in the attached file (comments).

Author Response
Dear reviewer#1:
T a point-to-point response.hanks for your kind comments, and we revised this manuscript with all your comments with

Reviewer 2 Report
Comments and Suggestions for Authors
This paper examines the fitness and predatory efficacy of four predatory mite species on two-spotted mites in high-altitude areas in China. The authors have conducted extensive laboratory work to assess the potential of these predatory mites. Despite this, there are several areas where the paper could be improved for clarity and scientific rigor:
In the Simple Summary, statements such as "Taken together, A. swirskii was considered the most effective predatory mite species to control the two-spotted mites in Tibet" (Lines 24-25) should be re-evaluated. This claim seems premature without field or even semi-field test results presented in the manuscript. Please also revise the title, the tile now looks like the authors did field tests. Additionally, the title could be misconstrued as implying that field tests were conducted. I recommend revising the title to reflect the scope of the laboratory studies presented more accurately.
Lines 100-102, please delete results in the introduction section
.
Line 109, please specify " controlled conditions".
In the Materials and Methods section (Lines 121-122), the rationale for selecting a temperature of 25.5°C and relative humidity of 80% for these assays is unclear, especially given the regional humidity data reported by Wang et al., 2023 (https://doi.org/10.1016/j.jtherbio.2023.103493). If the study's humidity levels were higher than the natural conditions in Lhasa, the relevance of these results to field applications might be questionable.
The Discussion section could benefit from a more direct comparison with previously published work. Consider removing or integrating the second paragraph into the Introduction to provide a clearer context for your findings.
Lastly, comparing fecundity data (Lines 378 to 384) with reference [44] should account for the different laboratory conditions, specifically the relative humidity levels used. It is crucial to control for these variables when drawing conclusions about species' performance at different altitudes. For example, if multiple factors are different, the authors should not make the statement regarding which species can survive better at high altitudes compared to lower altitudes.
Author Response
Dear reviewer#2:
Thanks for your kind comments, and we revised this manuscript with all your comments with a point-to-point response.

Reviewer 3 Report
Comments and Suggestions for Authors
All comments are included in the attached document.

It is acceptable and can be improved, mainly by changing passive sentences to active writing.
Author Response
Dear reviewer#3:
Thanks for your kind comments, and we revised this manuscript with all your comments with a point-to-point response.

Round 2
Reviewer 2 Report
Comments and Suggestions for Authors
The authors addressed my comments well. It has been sufficiently improved for publishing.
Author Response
Dear Reviewer#2:
Thanks a lot for your kind comments, the article improved with your hard work.
Best wishes for you.
Reviewer 3 Report
Comments and Suggestions for Authors
The manuscript looks better, but there are still multiple points mentioned in the previous review that were not addressed. Make sure to read all the comments in the attached pdf and make the required modifications.

Author Response
Dear Reviewer#3:
The manuscript had been revised with your kind comment, mainly including: 1) adding the order and family names of all species for the first appearance; 2) adding F value and df of all analysis; 3) correcting all embarrassed words with your comments; 4) remaking figures. Revised manuscript was in the attachment.
Best wishes for you.
